# Thermodynamic and Dynamic Transitions and Interaction Aspects in Reorientation Dynamics of *Molecular Probe* in *Organic Compounds*: A Series of *1-alkanols* with *TEMPO*

**DOI:** 10.3390/ijms241814252

**Published:** 2023-09-18

**Authors:** Josef Bartoš, Helena Švajdlenková

**Affiliations:** 1Polymer Institute of SAS, 845 41 Bratislava, Slovakia; helena.svajdlenkova@savba.sk; 2Department of Nuclear Chemistry, Faculty of Natural Sciences, Comenius University, 842 15 Bratislava, Slovakia

**Keywords:** *1-alkanols*, *spin probe TEMPO*, ESR, thermodynamic transitions, viscosity, crossover transition, polarity, proticity

## Abstract

The spectral and dynamic properties of *2,2,6,6-tetramethyl-1-piperidinyloxy (TEMPO)* in a series of *1-alkanols* ranging from *methanol* to *1-decanol* over a temperature range 100–300 K were investigated by *electron spin resonance (ESR*). The main characteristic ESR temperatures connected with slow to fast motion regime transition; *T*_50G_ ‘s and *T*_X1_^fast^ ‘s are situated above the corresponding glass temperatures, *T*_g_, and for the *shorter members,* the *T*_50G_ ‘s lie above or close to melting point, *T*_m_, while the *longer ones* the *T*_50G_ < *T*_m_ relationship indicates that the *TEMPO molecules* are in the local disordered regions of the *crystalline media*. The *T*_50G_ ‘s and especially *T*_X1_^fast^ ‘s are compared with the dynamic crossover temperatures, *T*_X_^VISC^ = 8.72*M*^0.66^, as obtained by fitting the viscosity data in the liquid *n-alkanols* with the empirical power law. In particular, for *N*_C_ > 6, the *T*_X1_^fast^ ‘s lie rather close to the *T*_X_^VISC^ resembling *apolar n-alkanes* [PCCP 2018,*20*,11145-11151], while for *N*_C_ < 6, they are situated in the vicinity of *T*_m_. The absence of a coincidence for lower*1-alkanols* indicates that the *T*_50G_ is significantly influenced by the mutual interaction between the *polar TEMPO* and the *protic polar medium* due to the increased polarity and proticity destroyed by the larger-scale melting transition.

## 1. Introduction

In general, the dynamics of *glass-forming liquids*, i.e., *organics* and *inorganics* forming the supercooled *liquid* by their cooling below the melting temperature, *T*_m_, and finished via a *liquid*−to−*glass* transition to a *glass* below the glass temperature, *T*_g_, is non-monotonous and exhibits a change at the so-called dynamic crossover temperature, *T*_cross_ = *T*_B_ or *T*_X_, lying between *T*_m_ and *T*_g_ [1,2,3,4,5,6,7,8,9,10,11,12]. This dynamic crossover phenomenon between the relatively *weakly* and *strongly* changing supercooled *liquid dynamics* is observed using experimental techniques, such as *viscosity* (*VISC*) at *T*_B,η_ or *T*_X_ [1,2], and *dielectric spectroscopy* (*DS*) at *T*_B,DS_^ST^ [3,4] or *T*_B,DS_^MG^ [5], as well as *T*_B,DS_^KWW^ [6] and *T*_B,DS_^SCH^ [7].

Usually, the crossover temperatures are determined by fitting the supercooled liquid dynamics using a combination of classic phenomenological expressions for viscosity, *η*, or structural relaxation time, *τ*, such as the *Vogel–Fulcher–Tamman–Hesse* (*VFTH*) equation [1], or using the *power law* (*PL*) equation [2]. Lately, a special evaluation method using *Stickel’s* temperature-derivative analysis or the more general *Martinez–Garcia* apparent enthalpy analysis of the relevant dynamic quantities, giving *T*_B,DS_
^ST^ and *T*_B,DS_
^MG^, respectively, was proposed [3,4,5]. Other ways of determination of the crossover temperature are based on the onset of the increasing broadening of the frequency dispersion of the structural relaxation time distribution, and on the change in structural relaxation strength, Δ*ε_α_*, leading to *T*_B,DS_
^KWW^ or *T*_B,DS_
^SCH^, respectively [6,7]. In the case of the *PL* eq. with *T*_X_ [2,9], this expression is rationalized theoretically using the *idealized mode coupling theory (I*−*MCT*) of *liquid dynamics* [11] via the derivation of the same form of temperature dependence for viscosity and relaxation time using the so-called critical temperature, *T*_c_ ≈ *T*_X._

The crossover transition is a very significant feature of the supercooled liquid’s behavior, as has been demonstrated by the findings of several empirical correlations of *T*_B_ or *T*_X_ with various characteristic temperatures of a variety of structural–dynamic phenomena, such as the decoupling or bifurcation of the primary α relaxation and the secondary β process, *T*_αβ_, from *DS* or *dynamic light scattering* (*DLS*) [8]. Moreover, the crossover phenomenon is also illustrated by various *extrinsic* probe techniques, such as *FS, ESR* and *PALS*. These have revealed the decoupling of the translation from the rotation of *molecular probes* and *medium* dynamics at *T*_decoup_, either for the relatively large *fluorescence probes* via *fluorescence spectroscopy (FS)* [13] or the decoupling of rotation of the *spin probes* from the *medium* dynamics, using *electron spin resonance (ESR)* [14]. Finally, crossover in the supercooled liquid state is also manifested by a bend effect in *ortho-positronium* lifetime *τ*_3_ vs. *T* dependence, as detected by *positron annihilation lifetime (PALS)*. This slope change reflects a change in the free volume expansion at the characteristic PALS temperature *T*_b1_^L^ above *T*_g_ in *amorphous glass*−*formers* [15]. Evidently, it is increasingly recognized that the dynamic crossover before glass transition temperature plays an essential if not fundamental role in our understanding of the glass transition phenomenon [9].

In contrast to the aforementioned cases of *amorphous glass-formers*, observations of crossover transition in strongly crystallizing, i.e., relatively hardy supercooled *apolar* and *polar organics,* is substantially more difficult. This is connected with the problem of the formation of sufficiently large *amorphous* domains in the otherwise dominantly *ordered material,* and subsequently, with their characterization by suitable *experimental techniques*. Recently, we have proposed one *special method* for creating such *amorphous* domains in *crystalline materials,* consisting of the introduction of an appropriate *molecular probe* disordering the immediate surroundings of the otherwise *ordered medium*. This includes the *spin probe* (*2,2,6,6*−*tetramethyl piperidin*−*1*−*yl) oxyl (TEMPO)* with *V*_TEMPO_^vdW^ = 170 Å^3^ in a series of *apolar n*−*alkanes* ranging from *n-hexane* to *n*−*nonadecane* using the *ESR technique* [16]. On the basis of the close correlation between one of the characteristic ESR temperatures, namely, *T*_X1_^fast^, lying a bit above the main slow−to−fast transition at *T*_50G_, and marking the onset of the pure fast motion regime of *TEMPO* and the crossover temperatures, *T*_X_
^VISC^, as obtained from fitting the corresponding viscosity data for a series of *n-alkanes* using the PL eq., dynamic crossovers in the local disordered regions around the *probe molecules* in the otherwise dominantly *crystalline organics* were detected.

One of the most important aspects of extrinsic probe techniques such as *ESR* is the potential interaction between the *probe* used and the *medium’s constituents,* which can to a greater or lesser extent influence the corresponding *probe* response to the investigated *organic matrix.* In our previous work on a series of *apolar* crystallizing *n-alkanes*, we used one of the smallest *polar spin probes, TEMPO,* where in this interaction aspect is supposed to be small [16]. The aim of this work was to test other types of *strongly* crystallizing *organic media* consisting of *protic polar compounds*, such as *1*−*alkanols*, with the potential for an intermolecular *H*−bonding interaction, not only between its own *polar molecules* but also between these *polar molecules* and *polar spin probe TEMPO*. The spectral and dynamic data obtained for the *TEMPO* on the family of aliphatic *monoalcohols* or *1*−*alkanols H(CH_2_)_N_ OH* with *N*_C_ = 1–10, i.e., ranging from *methanol* to *1*−*decanol,* are interpreted using the newly analyzed viscosity data in the literature in order to reveal the roles of the thermodynamic and dynamic transitions, as well as of the interaction aspect, in the main slow-to-fast transition behavior of the *spin probe TEMPO* used.

## 2. Results and Discussion

### 2.1. Thermodynamic and Crossover Transition Behaviors in 1−alkanols

It is well known that *1*−*alkanols,* similarly to *n*−*alkanes,* belong to the class of relatively easily crystalizing organic *compounds*. For this strong ordering tendency, they must have special means of preparation of the *totally* or *partially amorphous samples,* with the one exception of *1*−*propanol* (*C3OH*), which is a very good *glass*−*former* [17].

#### 2.1.1. Thermodynamic Transitions in *1*−*alkanols*

Figure 1 and Figure 2 and Table 1 summarize the data from the literature on the three basic thermodynamic transitions of *condensed materials,* i.e., the *glass*−to−*liquid* (devitrification) transition of the *amorphous* phase, the *solid*-to-*liquid* (melting) transition of the *crystalline* phase, as well as the *liquid*−to−*gas* (evaporation) transition of the *liquid* phase to *gas*. In general, the corresponding transition temperatures, i.e., glass temperature *T*_g_ and melting temperature *T*_m_, for *1*−*alkanols,* exhibit a *non-monotonous* characteristic as a function of their molecular, size expressed by the number of carbon atoms in the chain, *N*_c_, or molecular weight, *M,* while the boiling temperature, *T*_b_, shows a *monotonous* type of dependence over the whole molecular size interval. In contrast to the *melting* with well-defined *T*_m_ values [18], the *T*_g_ values measured so far exhibit a scatter up to 10 K, which depends on both the preparation procedure and the measuring technique, such as *dynamic–mechanical spectroscopy (DMS)*, *differential thermal analysis (DTA)* or *differential scanning calorimetry (DSC)*, and *dielectric spectroscopy (DS)* [19,20,21,22,23,24,25,26,27,28], with this value apparently diminishing with increasing molecular size. In spite of this fact, the *T*_g_ value for the shortest *1*−*alkanols* decreases from *methanol* (*C1OH*) to *ethanol* (*C2OH*), followed by a *monotonous* increasing trend, starting from *C2OH* in the DMS data [26], or from *1*−*propanol* (*C3OH*) in the CAL data [17]. As originally proposed by *Faucher* and *Koleske* [26], the DMS results could be described by the power law (PL)-type expression as a function of molecular weight, *M*:*T*_g_ = A*M*
^α^(1)
where A and α are empirical parameters of a given homologous series of *compounds*. As seen in Figure 1A, they exhibit similar trends depending relatively strongly on the method of generation of the *amorphous material* and the set up used in *DMS* or *DTA*, respectively. The latter values of *T*_g_
^DTA^ coinciding with *T*_g_
^CAL^ in cases of lower *1-alkanols*, such as *ethanol*, *1*−*propanol* and *1*−*butanol,* as studied by the special CAL technique, i.e., *quasi-adiabatic calorimetry (QADC)* [17,28], are considered to be more reliable, mainly because of the experimental complexity of *DMS*.

As for the melting transition of *1*−*alkanols*, the same expression can be approximately used for the melting points:*T*_m_ = C*M*
^γ^(2)
where C = 8.41 and γ = 0.705 are empirical parameters of the melting of a given homologous series of *compounds,* as derived mainly from the calorimetric data [18] in Figure 2. A similar approach has recently been applied for *n*−*alkanes* and *monoalcohols* in spite of the very pronounced *zig-zag* effect for the *former,* with a similar γ value of 0.7, as given by *Novikov* and *Rössler* [29].

Finally, in Figure 1B, the estimated values of glass transition, *T*_g_*, calculated according to the well-known empirical rule for many *organic* and *inorganic glass-formers* (*T*_g_* = (2/3) × *T*_m_—see e.g., Refs. [29,30,31,32]), are also listed. Given their comparison with the measured *T*_g_ data from *DTA* or *DMS*, it follows that this rule is not valid for our series of the first ten *1-alkanols*, with the one exception of *C3OH*. Alternatively, the measured *T*_m_
^CAL^/*T*_g_
^DTA^ ratios fulfill the rather different empirical rule of ~1.70, instead of *T*_m_/*T*_g_* = (3/2) = 1.50, which is valid again for *C3OH* only—see Figure 3.

In the case of *T*_m_, the full horizontal lines represent the constant value of *T*_m_/*T*_g_ = 1.5 derived from the empirical rule, and the dotted line represents *T*_m_/*T*_g_~1.68 for our series of *1-alkanols,* whereas in the case *T*_X_, with one exception for *C1OH,* an increasing linear trend of *T*_X_/*T*_g_ is found. Finally, *T*_X_/*T*_m_~0.81.

#### 2.1.2. Dynamic Crossovers in *1-alkanols*

As mentioned in the introduction, the crossover temperature in the supercooled liquid state of many *organic compounds* can be obtained using the *power law* (*PL*) equation, connecting the viscosity of liquids, *η*, with temperature *T* in the *normal* liquid and *weakly* supercooled liquid states [2,9,16]. Figure 4, Figure 5 and Figure 6, as well as Table 1, give the results of this method of determination of the crossover temperature, *T_X_*. Thus, Figure 4 presents compilations of the viscosity data for a series of ten *1-alkanols* ranging from *methanol* (*C1OH*) up to *1-decanol* (*C10OH*) as a function of temperature, mostly derived from the two large summarizing literature sources [33,34]. Most viscosity data of *1-alkanols* fall into the *normal* liquid state between the melting temperature, *T*_m_, and the boiling temperature, *T*_b_. For the two lower *members* of this series, namely, *ethanol* (*C2OH*) and *1-propa nol (C3OH)*, the viscosities were also measured in the supercooled liquid state below the corresponding *T*_m_’s; these values were only slightly lower for *C2OH* [35], but in *C3OH* they almost reached down to the corresponding *T*_g,_ because of its very good glass-forming ability [17,35,36,37]. Moreover, the additional liquid data from ref. [38] are included.

All the viscosity data for the series of the first ten *1-alkanols* can be described by the *power law (PL)* equation:(3)$$\eta_{}\left(T\right)=\eta_{\infty }{\left[\frac{\left(T-T_{X}\right)}{T_{X}}\right]}^{-\mu }$$
where *η*_∞_ is the pre-exponential factor, *T*_X_ is the empirical characteristic dynamic *PL* temperature or the theoretical critical MCT temperature *T*_c_, and *μ* is a non-universal coefficient. The corresponding fitting curves are plotted in Figure 4 and the obtained crossover temperatures *T*_X_ are listed in Table 1 and Figure 6. In the above-mentioned case of *1-alkanols* for which viscosity data in the supercooled liquid state also exist [35,36,37], the *T*_X_ values for, e.g., *C2OH,* extracted from fitting over the usual *normal* liquid state ranging *T*_b_−*T*_m_ = 192 K, and over the whole accessible temperature range of 227 K [35] (*T*_b_−*T*_min_ = 227 K), change by 4 K only. Similarly, for the very good *glass-former C3OH,* this difference reaches 3 K, which is towards the lower value, as it also includes the *strongly* supercooled liquid range data from Ref. [35]—see Figure 5A,B. Thus, the *PL* equation fit over the *normal* liquid state appears to be a very good approximation only for obtaining the *T*_X_ values lying in the supercooled liquid state below the corresponding *T*_m_ values in *strongly* crystallizing organics, such as *1-alkanols*. These are also listed in Table 1, together with the few determinations based on the first three *members*, namely, *C1OH, C2OH* and *C3OH,* as given by other authors [2,5,9].
ijms-24-14252-t001_Table 1Table 1Basic physical properties of investigated *1-alkanols*.*1-alkanol**M*g/mol*T*_g_K*T*_X_K*T*_m_^a^K*T*_b_^a^K*A*_zz’_ (100K) ^b^G*A*_iso_ (RT) ^c^G*μ*_g_^d^D*μ*_l_^e^D*ε*_r_^d^-***MeOH***32.04108.2 ^f^ 110.2 ^g^ 103 ^h^ 104.2 ^i^135.8135^2^175.233837.8716.461.702.7033***EtOH***46.07103.2 ^f^ 100.2 ^g^98.4 ^i^ 96 ^j^ 97 ^k^111111^5^158.935136.8116.301.692.8125.3***1-PrOH***60.10108 ^f^ 98 ^j^ 109 ^g^100 ^i^ 103 ^l^ 110 ^m^126.5139^9^146.837136.1516.231.682.8720.8***1-BuOH***74.12119 ^f^ 111.7 ^i^113.6 ^n^ 111 ^o^ 119 ^m^142183.739135.9516.161.662.8817.8***1-PentOH***88.15120.1 ^f^ 124 ^g^120 ^i^ 126.1 ^l^ 127.4 ^m^168194.841135.916.051.702.9715.1***1-HxOH***102.18125 ^f^ 138 ^g^ 129.9 ^i^ 135 ^m^178224.443035.7515.911.652.8713.0***1-HptOH***116.20123 ^f^ 143 ^g^141.2 ^i^ 141.9 ^m^199.723944935.6015.881.712.9911.75***1-OctOH***130.23149 ^g^ 149.9 ^i^ 148.3 ^m^225.4258.146835.4515.831.682.9010.30***1-NonOH***144.25153 ^i^ 154.4 ^m^23626848735.2515.751.602.738.83***1-DecOH***158.28(160.1) ^m^23927950135.1515.701.602.707.93^a^ Ref. [18]; ^b^ uncertainty = 0.05 G; ^c^ uncertainty = 0.03 G; ^d^ Ref. [33]; ^e^ Ref. [39]; ^f^ Ref. [19]; ^g^ Ref. [20]; ^h^ Ref. [21]; ^i^ Ref. [22]; ^j^ Ref. [23]; ^k^ Ref. [24]; ^l^ Ref. [25]; ^m^ Ref. [26]; ^n^ Ref. [27]; ^o^ Ref. [28].

It is shown that the *PL* equation is valid for a large number of *organic molecular glass-formers* over rather higher temperature range [2,5,9] It is also known that the I-MCT also works very well for the relatively lower viscosity regime [11]. In reality, although the viscosity does not diverge at *T*_X_ ≈ *T*_c_, several analyses of the slightly supercooled and normal liquid dynamics in various *organic glass-formers* in terms of the *extended mode coupling theory (E-MCT)*, which removes this singularity, provide the *same* crossover temperature in the supercooled liquid phase [11,40]. Thus, the *T*_X_ parameter marks two distinct regimes of the *strongly* and *weakly* supercooled liquid dynamics [11]. In particular, it corresponds to the onset of dynamic heterogeneities, i.e., regions with slower dynamics embedded into regions of higher dynamics, when the decoupling of translation from rotation of the *molecular tracers* and the decoupling of rotation of the *molecular tracer* from that of the *medium constituents* occur, and the classic Stokes−Einstein or Debye−Stokes−Einstein laws, respectively, are violated [9,14].

Figure 6 displays the molecular size dependence of the extracted dynamic crossover temperature *T*_X_ for *1*−*alkanols*, together with its fitting curve, with a similar form to that of *T*_g_ and *T*_m_. Similar to the quantities of *T*_g_ and *T*_m_, after the initial decrease to the second lowest *member* the series of nine *1*−*alkanols,* the *power law* formula below is followed:*T*_X_ = B*M*
^*β*^(4)
with the *β* exponent, equaling 0.656, lying in between those for the glass temperature, *T*_g_(α = 0.503), and melting point, *T*_m_ (γ = 0.705).

Finally, returning to Figure 3, a comparison of *T*_X_/*T*_g_ vs. *T*_m_/*T*_g_ dependencies as a function of molecular size *N_C_* starting from *C2OH* shows a diametrically different trend for the former quantity with respect to the latter one, i.e., the increasing distance of the particular *T*_X_ from the respective *T*_g_ with the increasing molecular size of *1*−*alkanol*. This finding, together with the almost identical relative distance of *T*_X_ from *T*_m_ (ca. 0.81 × *T*_m_), indicates that the larger the molecule of *1*−*alkanol,* the larger the temperature range of the *strongly* (or *deeply)* (and correspondingly, the shorter the *weakly* (*slightly*)) supercooled liquid state. On the other hand, upon cooling the smaller *1*−*alkanols*, the *weakly* supercooled liquid state persists for longer, with a correspondingly shorter *deeply* supercooled liquid range.

### 2.2. ESR Data

#### 2.2.1. General Spectral and Dynamic Features

Figure 7 presents the 2*A*_zz’_ vs. *T* dependencies for our series of *spin systems: TEMPO*/*1-alkanols*. In all cases, the quasi-sigmoidal courses of these plots are found, with higher 2*A*_zz’_ values in a slow motional regime at relatively lower temperatures and the lower 2*A*_zz’_ ones in a fast motion regime in relatively higher temperature regions. The most pronounced feature of 2*A*_zz’_ vs. *T* dependencies is a more or less sharp change at the main characteristic ESR temperature, *T*_50G_ at which the 2*A*_zz’_(*T*_50G_) value reaches just 50 Gauss, corresponding to the correlation time of the *TEMPO* in a typical *organic medium* around a few ns. Note that the detailed spectral simulations of the *TEMPO* dynamics in several *organics*, including one of the investigated *1*−*alkanols*, namely, *1*−*propanol* [41], reveal that the *spin probe* population even at *T*_50G_ is not completely situated in the *fast* motion regime, which occurs at a slightly higher temperature, *T*_X1_^fast^. In addition to these main characteristic ESR temperatures, *T*_50G_ and *T*_X1_^fast^, other effects appear at *T*_X1_^slow^ and *T*_X2_^fast^, a discussion of which goes beyond the scope of this work, and will therefore be addressed elsewhere. All the 2*A*_zz’_ vs. *T* plots also include the afore-mentioned thermodynamic and dynamic temperatures: *T*_g_, *T*_m_ or *T*_X_, respectively. The mutual relationships of these three basic characteristic thermodynamic and dynamic temperatures with *T*_50G_ and *T*_X1_^fast^ in a series of *1*−*alkanols* will be discussed below, in Section 2.2.2.

In principle, the main slow-to-fast motion transition of the *TEMPO* in any *organics* is related not only to these thermodynamic and dynamic transitions, but it may also be influenced by further factors, such as the potential mutual interaction of a *polar spin probe* with *organic media*, especially *polar ones*. The values of anisotropic hyperfine constants *A*_zz’_ (100 K) at the lowest measured temperature of 100 K, and of isotropic ones *A_iso_* (RT) at room temperature, are summarized in Table 1. Their dependencies on *N*_C,_ as well as on some relevant *bulk* properties of the *media,* such as the *bulk* polarity of *media,* through their dielectric constant, *ε*_r_, will be discussed in the Section 2.2.3.

Finally, the mutual connections between the temperature parameters of the slow-to-fast transition, and the thermodynamic as well as dynamic ones, in relation to the polarization interaction of the *polar TEMPO probe* with a series of *polar 1*−*alkanols*, are discussed in Section 2.2.4.

#### 2.2.2. The Mutual Relationships of *T*_50G_ and *T*_X1_^fast^ with Thermodynamic and Dynamic Transitions

In Figure 8, global comparisons of the characteristic ESR temperatures *T*_50G_ and *T*_X1_^fast^ with the aforementioned thermodynamic and dynamic temperatures *T*_g_, *T*_X_ and *T*_m_ are presented. In all the cases, the slow-to-fast transition in all *1*−*alkanols* occurs above the corresponding *glass*−to−*liquid* transition, *T*_g_, i.e., in the amorphous phase of *liquid sample* or in the *local* amorphous *liquid* zones of *partially crystalline matrices*, at least. Figure 9 expresses these comparisons in terms of the corresponding ratios: *T*_50G_/*T*_m_, *T*_50G_/*T*_X_ and *T*_X1_^fast^/*T*_m_, *T*_X1_^fast^/*T*_X_. We can approximately distinguish two distinct regions of these ratios with a boundary at around *C5OH*
low *M* region: *C1OH-C5OH* with *T_50G_/T_m_* ≈ 1(5)
high *M* region: *C6OH-C10OH* with *T_50G_/T_X_* ≈ 1(6)

So, for higher *members* starting at *C6OH* to *C10OH,* with a relatively longer aliphatic part, we observe a plausible closeness between the characteristic ESR temperatures and the *T*_X_’s, indicating that the main ESR transition is related to the dynamic crossover between the *deeply* and *slightly supercooled liquid* state. This basic finding is similar to the previous one for a series of *n*−*alkanes* [16], with the fact that the *T*_X_’s of *1*−*alkanols* are higher than the *T*_X_ values for the corresponding *n-alkanes* with the same number of *C atoms* in the molecule. This difference indicates that the *spin probe TEMPO* is not fully surrounded by the *apolar aliphatic parts* of the *1*−*alkanol* molecules*,* and that the *polar* -*OH groups* influence its dynamics, as will be discussed later in Section 2.2.3. This indicates that the immediate environment of the molecular-sized *spin probe TEMPO* is locally disordered, and subsequently, sensitive to the crossover transition in this *local* amorphous phase. On the other hand, for low *M members* from *C1OH* to *C5OH,* the *T*_50G_ and *T*_X1_^fast^ values lie significantly above the corresponding *T*_X_ values, and they are situated in the vicinity of the corresponding melting temperatures, *T*_m_. This indicates that the slow-to−fast transition of *TEMPO* appears to be related to the global disordering process connected with the *solid*−to−*liquid* state phase transition in the otherwise *partially crystallized samples.*

#### 2.2.3. Isotropic and Anisotropic Hyperfine Constants *A*_iso_ (RT) and *A*_zz’_ (100 K) as a Function of the *N*_C_, Polarity and Proticity of *1*−*alkanols*

Figure 10 displays the anisotropic hyperfine constant, *A*_zz’_ (100K), and the isotropic. hyperfine constant, *A*_iso_(RT), of the *TEMPO* as a function of the chain length in the series of *1-alkanols* studied. Our values of *A*_iso_ (RT) for *TEMPO* are quite consistent with the few obtained for lower *members* of our series, namely, *C1OH–C4OH* [42,43,44]. Although both the quantities decrease with increasing chain size, a significant difference can be found in the corresponding trends. The former quantity shows two clear regions of distinct behavior: a sharper decreasing trend for low-*M members*, and a weaker one for higher *M members* above *N*_C_~4. On the other hand, the *A*_iso_ (RT) parameter is slightly reduced with a suggestion of a slight change at *N*_C_~6 as the number of *C atoms* in the molecule, *N*_C_, increases.

These basic empirical findings can be discussed in relation to the polarity of a set of *polar media*, with the dissolved *polar spin probe TEMPO μ*_TEMPO_~3 D [45], from both the phenomenological and theoretical viewpoints. First, the *A*_iso_(RT) values can be related to various measures of the polarity of the *medium*, e.g., the dipole moment of the *medium‘s molecule*, *μ*_entity,x_, as a measure of the polarity of the *individual entity* in a given phase state x = gaseous or liquid state or the static dielectric constant of the *medium*, *ε*_r_ (RT), as a measure of the polarity of the *bulk* liquid *medium*, as listed in Table 1. In the first case, evidently, no relationship can be found due to the quasi-constant values of the *gaseous*-phase *μ*_g_ = 1.66 ± 0.05 D [34] or the *liquid*-phase *μ*_l_ = 2.84 ± 0.15 D dipole moments [39]. On the other hand, Figure 11 displays the mutual relationships between the isotropic hyperfine constant, *A*_iso_ (RT), and the dielectric constant, *ε*_r_ (RT), of *1*−*alkanols* [34], together with those of the latter quantity at RT as a function of the number of *C atoms* in the chain inserted. Both the quantities decrease with *N*_C_, resulting in an *A*_iso_ (RT) vs. ε_r_ (RT) relationship, with approximately two regions showing distinct behavior: (i) for the lower *polar 1*−*alkanols* (*C10OH-C5OH*) with *ε*_r_ < ~17, with a strong sensitivity of *A*_iso_ (RT) to polarity and a weak one to proticity, and (ii) for higher *polar 1*−*alkanols* (*C3OH-C1OH*) with *ε*_r_ > ~17, with the weak sensitivity of *A*_iso_ (RT) to polarity and a stronger sensitivity to proticity, due to the increased population of *HO-groups* potentially interacting with the *spin probe TEMPO molecule*. The apparent boundary between both regions occurs at *N*_C_ = 4–5, i.e., for *1*−*butanol* or *1*−*pentanol*, where the *ε*_r_ (RT) vs. *N*_C_ (RT) plot changes rather notably from a sharply decreasing dependence to a slightly decreasing one, and where, at the same time, conformational degrees of freedom and related enhanced alignments of the *apolar* parts of the molecules start to occur. Interestingly, in spite of the absence of *ε*_r_ (100 K) data facilitating their direct comparison with *A*_zz’_ (100K), the boundary for this quantity seems to be consistent with that for *A*_iso_ (RT), suggesting a significant role of polarity and proticity in both the mobility states of the *spin probe TEMPO*. These findings of the solvent dependence of the different ESR parameters are consistent with the previous ones for *A*_iso_ (RT) [46,47], as well as for *A*_zz’_ (77 K) [48,49].

Our basic finding is similar to that derived for another larger nitroxide spin probe, *1*−*oxyl*−*2,2,5,5*−*tetramethyl pyrroline*−*3*−*methyl)methanethiosulfonate (MTSSL)*, in a series of 17 *solvents* ranging from *apolar methylbenzene (toluene)* (ε_r_ (RT) = 2.4) to highly *polar water* (ε_r_ (RT) = 80.4) and even more polar *formamid* (ε_r_ (RT) = 109), including most of the *members* of our *1*−*alkanol* series, with one exception for *1*−*pentanol* [50]. These authors similarly distinguished the following two regions, i.e., an “*apolar*”region for ε_r_(RT) < 25, where the sensitivity of *A*_iso_(RT) and *A*_zz’_(77K) to the polarity expressed by ε_r_ (RT) is large, and a “*polar*” region for ε_r_ (RT) > 25, where the sensitivity of *A*_iso_ (RT) and *A*_zz’_ (77 K) to the polarity is small, and the change is ascribed to the medium proticity.

However, it is evident that this division is rather arbitrary and very rough, because the former “*apolar*” region also includes many of our *polar 1*−*alkanols*. In connection with the aforementioned empirical relation between spectral parameters and *bulk* polarity, more elaborate theoretical approaches, based on models using the *medium* as a dielectric *continuum* with dielectric constant, *ε*_r_, and the *molecular solute*, e.g., *polar spin probe,* as a molecular entity localized in a spherical cavity [47,51], can be discussed. Within the reaction field concept of the polarization of the *continuum medium* by the *polar solute*, one obtains for the *Onsager’s* reaction field [52] and *Böttcher’s* reaction field [53] the following functional relations: *A*_iso_ = f [(*ε*_r_ − 1)/(*ε*_r_ + 1)] [47], or *A*_iso_ = f [(2*ε*_r_ + 1)/(2*ε*_r_ + *n*_D_^2^)], where *n*_D_ is the refraction index of the pure *nitroxide* [51]. Figure 12 displays a test of the validity of the first functional dependence for two basic groups of *organic compounds* at *RT* doped by *TEMPO*. The first is represented by a series of *apolar* and *aprotic polar solvents*, which range from *apolar benzene (BZ)* with *ε*_r_ (RT) = 2.3, to a highly *polar* but *aprotic dimethyl sulfoxide (DMSO)* with *ε*_r_ (RT) = 48.9, as taken from Ref. [50]. The other group, including our series of ten *1*−*alkanols* from *methanol* with *ε*_r_ (RT) = 33 to *1*−*decanol* with *ε*_r_(RT) = 7.9, differs significantly from the predicted linear trend due to the specific *protic* character of the *molecules*, allowing for *H*-bond formation between the *polar spin probe TEMPO molecule* and the *alkanol’s one*. This is quite consistent with the maximal value of *A*_iso_ (RT) = 17 Gauss [44] for highly *polar* and *protic water, where ε*_r_ (RT) = 80.4 [50]. The relatively large difference between *water* and the first *member of the alkanol family* is determined on the basis of theoretical calculations using density functional theory (DFT) interpreted in terms of the complexation of *nitroxide* with two *water* or one *methanol molecules*, respectively [45,50]. Moreover, a closer inspection of this group of *protic polar compounds* confirms the distinction of a series of *1*−*alkanols* into two subgroups, with distinct slopes of *A*_iso_ (RT) as a function of the corresponding dielectric function: (i) weaker for higher *members* from *C10OH* to *C6OH,* and (ii) stronger for shorter ones from *C5OH* to *C1OH,* with an approximate boundary between *C5OH* and *C6OH*, i.e., for *ε*_r_ (RT)~16.5. A similar situation can be found for the *Böttcher* reaction field due to the linearity between the respective functional forms. Both these findings appear to be consistent with the empirically determined boundary at *C4OH*-*C5OH,* as seen from the *A*_iso_ (RT) vs. *ε*_r_ (RT) plot *without* the inclusion of the polarization interaction between the *polar solute* and the *solvent,* as shown in Figure 11.

#### 2.2.4. Connection of the Main Slow-to-Fast Motion Transition of the *Spin Probe TEMPO* with the Polarity, Proticity and Thermodynamic and Dynamic Transition Behaviors of *1*−*alkanols*

In Figure 8 and Figure 9 in Section 2.2.2, we compare the characteristic ESR temperatures *T*_50G_ and *T*_X1_^fast^ of the slow-to-fast transition of *TEMPO* in a series of *1*−*alkanols* with the dynamic crossover *T*_X_ and thermodynamic transition temperatures *T*_m_, and also shown their mutual ratios as a function of the molecular size, *N*_C_, of the *media*. In particular, we revealed a step-like change in the main *spin probe TEMPO* transition from that seen at the dynamic crossovers at around *T*_X_ for the longer *chains*, to that related to thermodynamic transitions at around *T*_m_ for the shorter *molecules*, at *N*_C_~5.

Next, in Figure 10, Figure 11 and Figure 12 in Section 2.2.3, we present the relations of spectral parameters *A*_iso_ (RT) and *A*_zz’_ (100 K) to *N*_C_, as well as their phenomenological and theoretical relationships, especially for that of *A*_iso_ (RT) to the polarity properties of a set of *1*−*alkanols*. Here, we have observed a change in the trend of hyperfine interactions with the polarity and proticity of *1-alkanol media* at *N*_C_~4. Now, a combination of these findings indicates that the slow−to-fast transition in the mobility of *TEMPO* in a series of *1-alkanols* is relatively strongly dependent on the strength of intermolecular interactions between the *polar* constituents of the *polar media*, and between the *polar spin probe* and the polarity and proticity of the *1*−*alkanols* investigated. In the longer *members* of the *1-alkanol family,* with the relatively higher population of *apolar* aliphatic *methylene groups* related to a weakly changing polarity, the slow-to-fast transition is related mainly to the dynamic crossover process around *T*_X_, similar to what is seen for the *apolar n*−*alkanes* [16]. On the other hand, in the shorter *members* with relatively higher dielectric constants and proticity due to the relatively higher populations in the *polar hydroxyl groups*, a larger-scale disorder process connected with the *solid*-to-*liquid* phase transition around *T*_m_ is needed in order to destroy not only the dense *H*-bonding *network* between the *medium’s molecules*, but also to destroy the clustering of *polar TEMPO molecules* with them, and subsequently, and the appearance of slow−to−fast transition in the mobility of *TEMPO.* The critical molecular size of *1*−*alkanol* for this step-like change in the slow-to-fast transition of *TEMPO* lies at *N*_C_~5, below which the polarity and proticity aspects of the *media* become dominating factors.

## 3. Experimental

### 3.1. Materials and Methods

A series of *1*−*alkanols* ranging from *methanol* to *1*−*decanol*, received from Sigma-Aldrich, Inc., St. Louis, MO, USA, was used as the model *protic polar media*. Our choice of this series stems from the fact that all of the used *1*−*alkanols* are in the liquid state at room temperature, making *spin probe system* preparation easier. As an *extrinsic* particle, the *spin probe 2,2,6,6*−*tetramethyl*−*1*−*piperidinyloxy (TEMPO)* with a quasi-spherical shape and a relatively high dipole moment of *μ*_TEMPO_ = 3 D [45] was applied in the *deoxygenated 1*−*alkanols* at a very low concentration of ~5 × 10^−4^ M.

### 3.2. ESR

ESR measurements of the very diluted *spin systems 1-alkanol*/*TEMPO* were performed on the X-band Bruker–ER 200 SRL spectrometer operating at 9.4 GHz with a Bruker BVT 100 temperature variation controller unit. ESR spectra were recorded after cooling at a rate ~− 4 K/min in a heating mode over a wide temperature range from 100 K up to 300 K, with steps of 5–10 K. To reach the thermal equilibrium, the sample was kept at a given temperature for 10 min before starting *three* spectra collections. The temperature stability was ± 0.5 K. The microwave power and the amplitude of field modulation were optimized to avoid signal distortion. The ESR spectra were evaluated in terms of the spectral parameter of mobility, 2*A*_zz’_ (*T*), i.e., the z-component of the anisotropic tensor of the hyperfine interaction *A*(*T*), corresponding to the outermost peak separation of the triplet spectra of the *spin probe* of *nitroxide* type in a given *medium*, as a function of temperature and the subsequent determination of the spectral *T*_50G_ parameter [50,54]. This is the characteristic ESR temperature, at which 2*A*_zz’_ reaches the conventional value of 50 Gauss (G). Additional characteristic ESR temperatures can be obtained, and used to describe in detail the slow-to-fast regime transition zones over more or less wide temperature intervals around *T*_50G_, as well as in both slow and fast motion regimes: *T*_Xi_
^slow^, *T*_Xi_
^fast^ [55,56,57]. In addition to the anisotropic hyperfine splitting parameter, *A*_zz’_ (100 K), the isotropic hyperfine constant values, *A*_iso_ (RT), of the *spin probe TEMPO,* as a measure of its interaction with a given *medium*, were determined at room temperature under the fast motion condition in a low-viscosity *media* [50].

## 4. Conclusions

The spectral and dynamic behaviors of the *spin probe TEMPO* in a series of *1*−*alkanols* ranging from *methanol* to *1*−*decanol*, over a wide temperature range from 100 K up to 300 K, using *electron spin resonance (ESR*) are reported. For all of the *alkanols*, the main characteristic ESR temperatures connected with slow-to-fast motion regime transition, namely, *T*_50_ and *T*_X1_^fast^, are situated above the corresponding glass temperatures, *T*_g_, and for the first five shorter *members,* the *T*_50G_ values lie in the vicinity of the melting point, *T*_m_, while for the *longer ones*, the *T*_50G_ < *T*_m_ relationship indicates that the *TEMPO molecules* are in the local disordered regions of the *crystalline media*. The *T*_X1_^fast^ values are compared with the dynamic crossover temperatures, parametrized as *T*_X_^VISC^ = 8.72*M*^0.66^, which were obtained by fitting the viscosity data for the liquid *1*−*alkanols* with the empirical power law. In particular, for *N*_C_ = 6–10, the *T*_X1_^fast^ values lie relatively close to the *T*_X_^VISC^ seen for *apolar n-alkanes*, while for *N*_C_ = 1–5, they are situated *above* the respective *T*_X_^VISC^ values in the vicinity of *T*_m_. The absence of such a coincidence for lower *1*−*alkanols* indicates that the slow-to-fast motion transition is significantly influenced by the mutual interaction between the *polar TEMPO* and the *protic polar medium,* due to the increased polarity and proticity, which are destroyed at higher temperatures associated with the larger-scale *solid*-to-*liquid* transition.

## Figures and Tables

**Figure 1 ijms-24-14252-f001:**
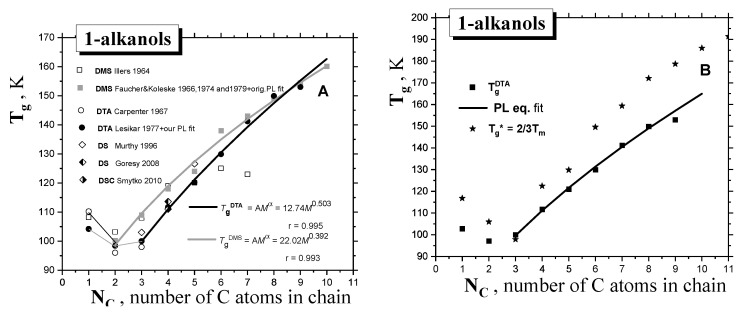
(**A**) *Glass*−to−*liquid* transition temperature *T*_g_ of *1*−*alkanols* as a function of the molecular size expressed by the number of *C atoms* in the *chain*, *N*_C_. Two fits of the *T*_g_ values from the *DMS* [20,26] and *DTA* data sets [22] via the *PL* equation (*T*_g_ = A*M*
^α^) are included, (**B**) Comparison of the *T*_g_
^DTA^ values with the empirical rule: *T*_g_* = (2/3) *T*_m_.

**Figure 2 ijms-24-14252-f002:**
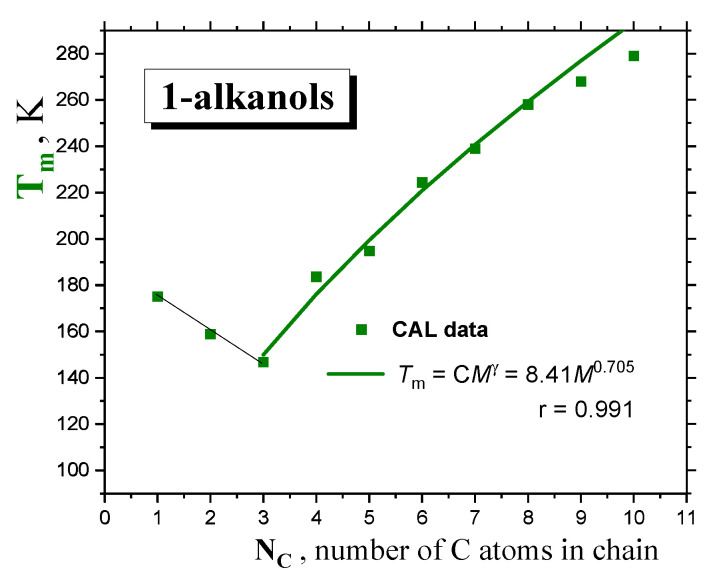
The melting temperature *T*_m_ of *1-alkanols* as a function of the molecular size expressed by the number of *C atoms* in the *chain, N*_C_. Fit of the *T*_m_’s from the *CAL* data set from Ref. [18] via the *PL* equation *T*_m_ = C*M*
^γ^ is included.

**Figure 3 ijms-24-14252-f003:**
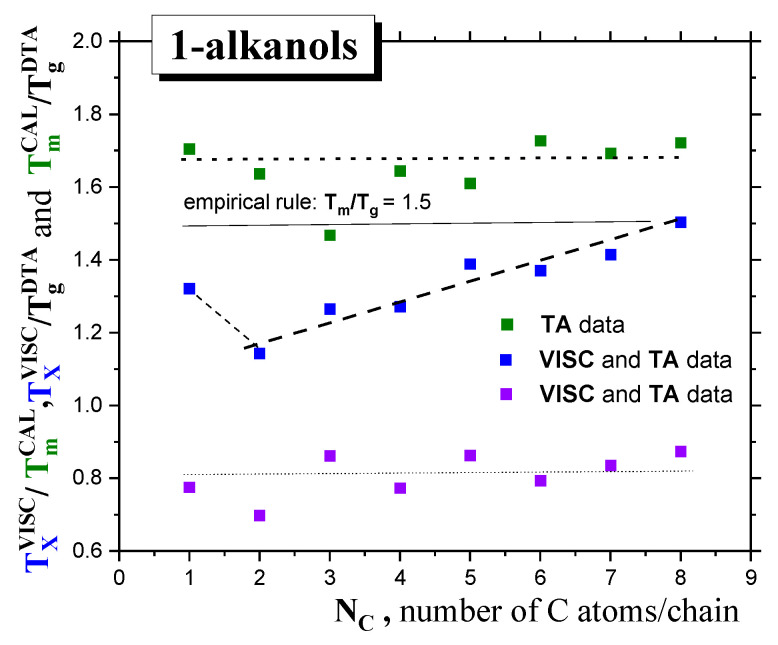
The *T*_m_/*T*_g_, *T*_X_/*T*_g_ and *T*_m_/*T*_X_ ratios of *1-alkanols* as a function of the molecular size expressed by the number of *C atoms* in the *chain, N*_C_.

**Figure 4 ijms-24-14252-f004:**
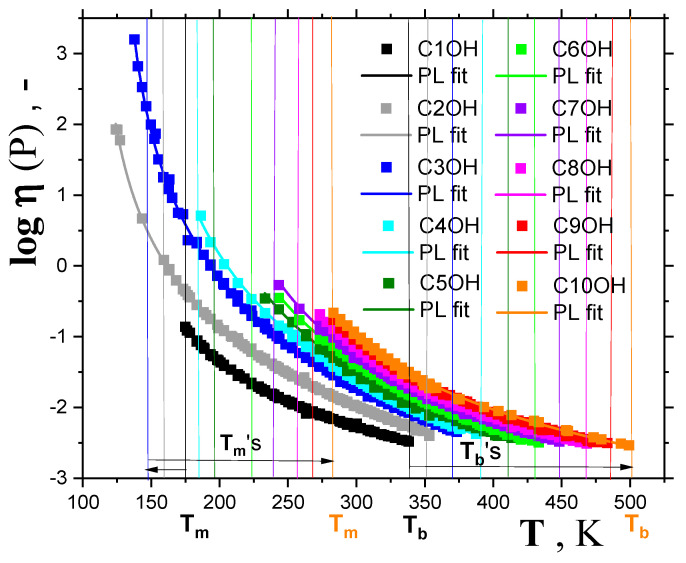
Viscosities for a series of *1*–*alkanols* as a function of temperature together with the respective *power law* (PL) fittings given by Equation (3). The vertical lines in the same colors as the corresponding experimental points for each *member* of a series of *1*–*alkanols* mark the melting temperatures (on the left) and the boiling points (on the right), with two extrema demonstrations for *methanol* (black points and lines) and *1*–*decanol* (orange points and lines). Trends in *T*_m_ and *T*_b_ are shown by the two arrows at the bottom of the plot.

**Figure 5 ijms-24-14252-f005:**
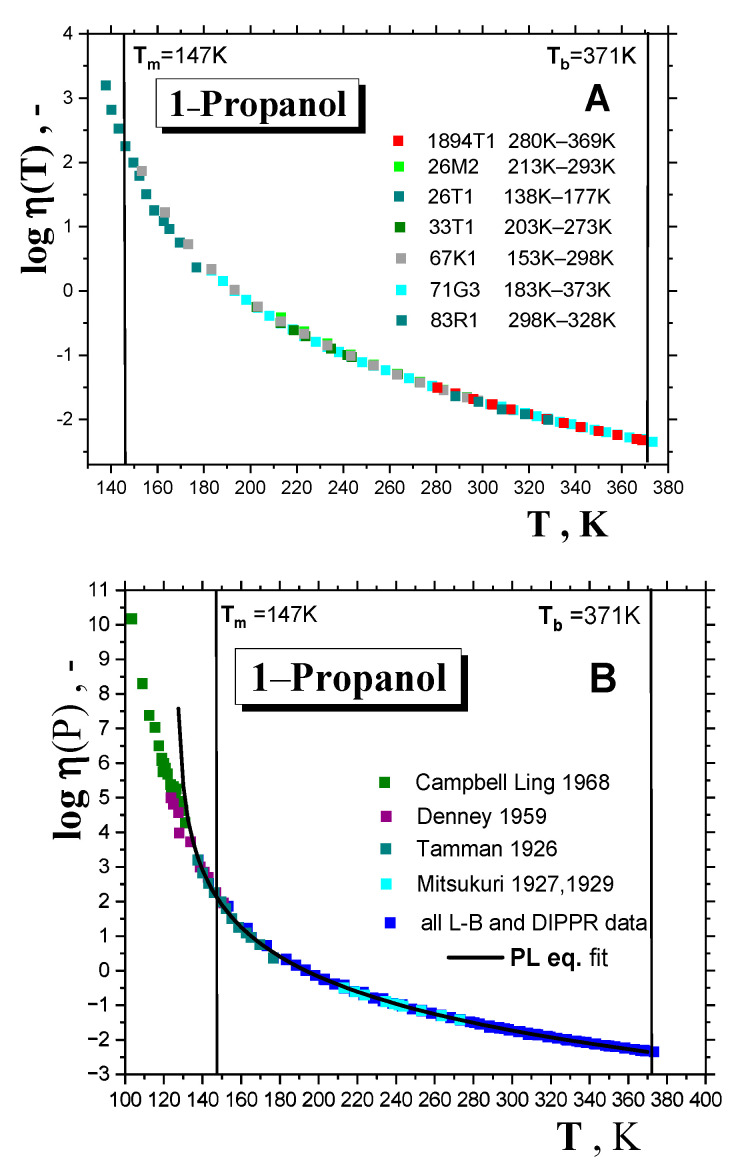
Viscosity of a good *glass*–*former 1*–*propanol* over (**A**) a restricted temperature range from *T*_b_ down to slightly below *T*_m_ using the data sets summarized in Ref. [33] (*Landolt–Börnstein* Table data) and Ref. [34] (*DIPPR* data), and (**B**) an extraordinarily wide temperature range from *T*_b_ down almost to *T*_g_ with the addition data from Refs. [35,36,37] in the *strongly* supercooled liquid state, as well as from Ref. [38] in the liquid one, together with the best *PL* equation fitting. The original references marked, such as 1891T1 and 26M2, can be found in Ref. [33].

**Figure 6 ijms-24-14252-f006:**
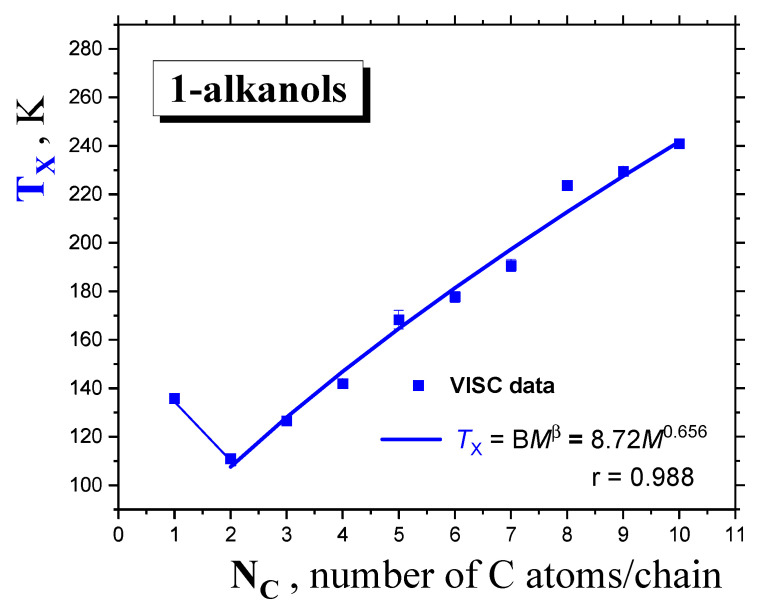
Dynamic crossover temperature *T*_X_ of *1*−*alkanols* as a function of the molecular size expressed by the number of *C atoms* in the *chain N*_C_. The fit of the *T*_X_ values from VISC data via the *PL* equation of the form *T*_X_ = B*M*
^β^ is included.

**Figure 7 ijms-24-14252-f007:**
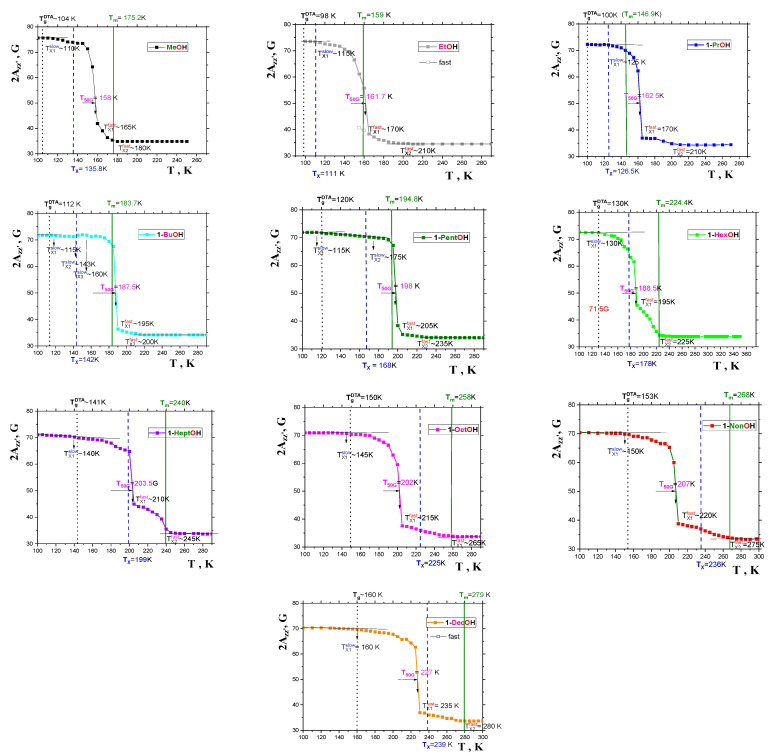
Spectral parameter 2*A*_zz’_ in a series of *1*−*alkanols* as a function of temperature. The colors for the ESR data for the individual *1*−*alkanols* are the same as for their viscosity data in Figure 4. The characteristic ESR temperatures *T*_Xi_
^slow^, *T*_50G_ and *T*_Xi_
^fast^ are marked, and the thermodynamic temperatures *T*_g_^DTA^ and *T*_m_^CAL^, as well as the dynamic one, *T*_X_, are depicted by the black, olive and blue lines, consistently with Figure 1, Figure 2 and Figure 6 and are discussed in the text in detail.

**Figure 8 ijms-24-14252-f008:**
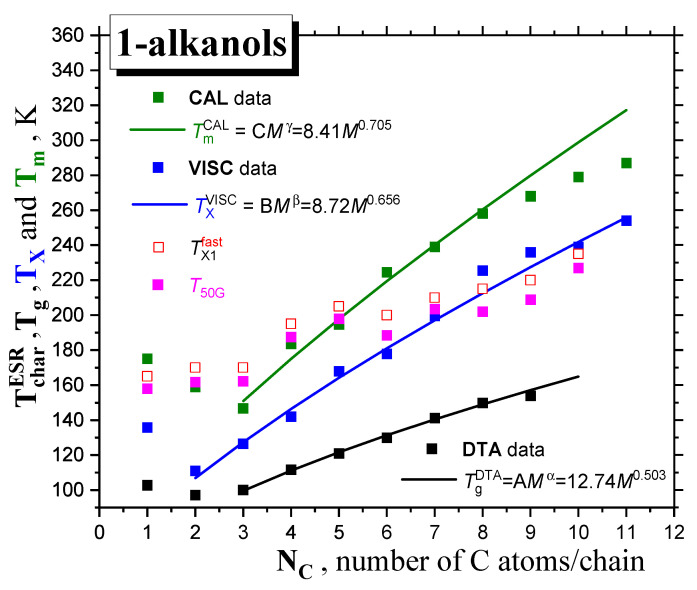
Comparison of the characteristic ESR temperatures, *T*_50G_ and *T*_X1_^fast^, with the thermodynamic and dynamic temperatures *T*_g_ ^DTA^, *T*_X_
^VISC^ and *T*_m_
^CAL^, together with their corresponding *PL* fits.

**Figure 9 ijms-24-14252-f009:**
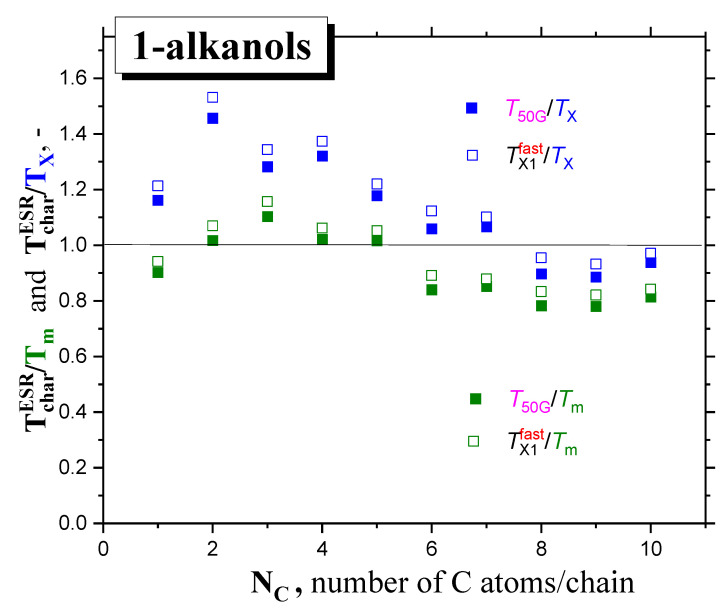
Comparison of the ratios of the characteristic ESR temperatures, *T*_50G_ and *T*_X1_^fast^, with the dynamic and thermodynamic temperatures, *T*_X_ or *T*_m_, respectively, as a function of *N*_C_.

**Figure 10 ijms-24-14252-f010:**
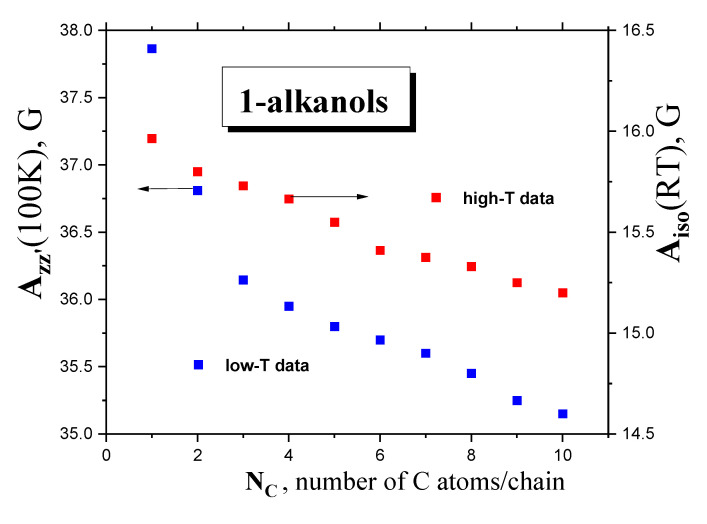
Hyperfine constants of *TEMPO A*_zz’_(100 K) and *A*_iso_ (RT) in a series of ten *1*−*alkanols* as a function of *N*_C_.

**Figure 11 ijms-24-14252-f011:**
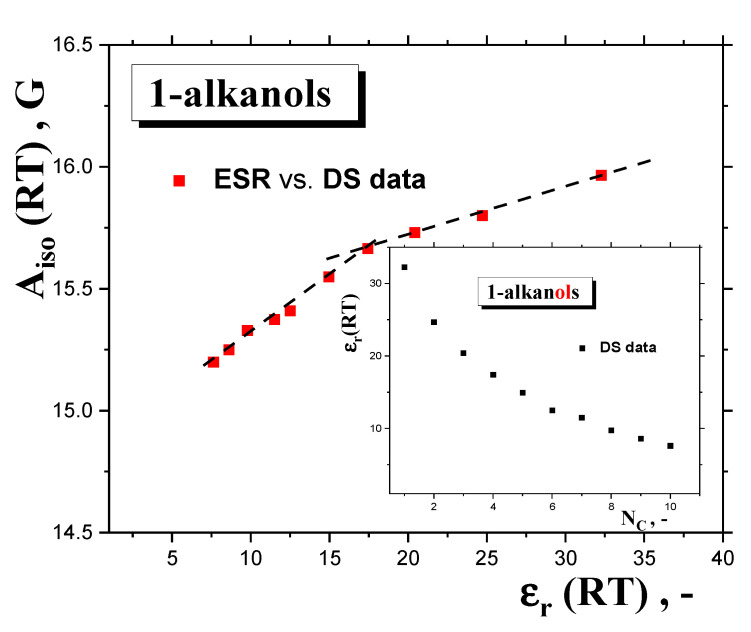
Empirical relationship between the isotropic hyperfine constant, *A*_iso_ (RT), and the relative permitivity, *ε*_r_ (RT), of *1*−*alkanols*. Insert contains the latter quantity at RT from Table 1 as a function of *N*_C_.

**Figure 12 ijms-24-14252-f012:**
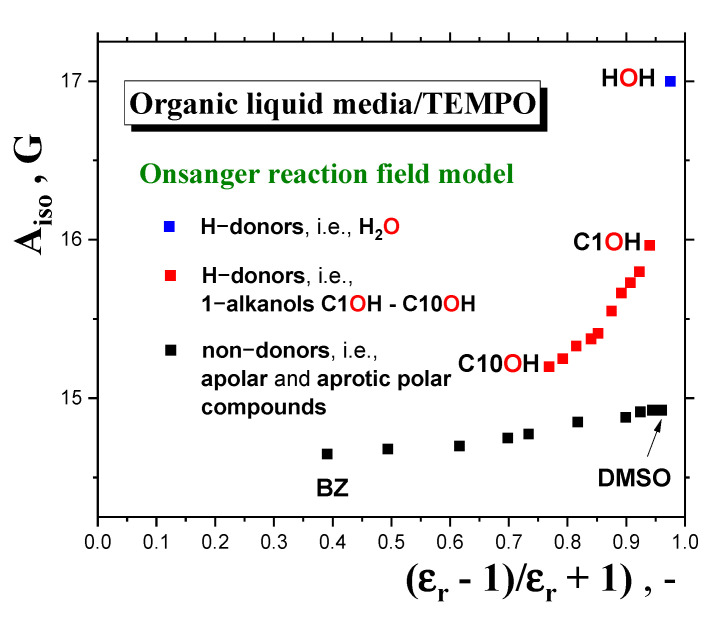
Test of the *Griffith−Onsanger* model for isotropic hyperfine constant, *A*_iso_ (RT), as a function of the polarization expression ε_r_ (RT) – 1)/(ε_r_ (RT) + 1) of the *Onsanger* reaction field model for three types of media: *apolar*, such as *benzene (BZ)* [44]; *aprotic polar*, such as *dimethylsulfoxide (DMSO)* [44]’ and *protic polar* compounds, such as *water* [44] and our series of ten *1*−*alkanols*.

## Data Availability

Not applicable.

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
