# Peer review of "Thermodynamic and Dynamic Transitions and Interaction Aspects in Reorientation Dynamics of Molecular Probe in Organic Compounds: A Series of 1-alkanols with TEMPO"

_ijms, 2023, doi:10.3390/ijms241814252_

Round 1

Reviewer 1 Report

Good paper. Some improvements may be done. Please put all equations in M&M section. Figure 8, there are some point out of line that could be excluded. Figure 7 may be improved, please include letters such as A,B,C... and discuss each figure in the main text. There are large paragraphs such as the first one in introduction. Please separate in 10 lines maximum. 

Good English language. From my side, no major changes are required.

Author Response

Our response to Reviewer 1 is in the attached file.

Reviewer 2 Report

Current manuscript entitled “Thermodynamic and dynamic transitions and interaction aspects in reorientation dynamics of molecular probe in organic compounds: A series of 1-alkanols with TEMPO” seems good and can be accepted after addressing the following comments.

1.      In the introduction compare the work with the existing literature.

2.      How much is the atmospheric pressure when performing the experiments

3.      Key findings of the performed study must be included in the revised version.

4.      Improve the image quality of the figures.

5.      Clear statements of the novelty of the work should also appear briefly in the Abstract and Conclusions sections.

6.      The manuscript has grammatical errors/ typos/ incomplete sentences and non-relative phrases.

Minor editing of English language required

Author Response

Our response to Reviewer2 is in the attachment.
